# HER3 Alterations in Cancer and Potential Clinical Implications

**DOI:** 10.3390/cancers14246174

**Published:** 2022-12-14

**Authors:** Mary Kate Kilroy, SoYoung Park, Wasim Feroz, Hima Patel, Rosalin Mishra, Samar Alanazi, Joan T. Garrett

**Affiliations:** 1Department of Pharmaceutical Sciences, James L. Winkle College of Pharmacy, University of Cincinnati, Cincinnati, OH 45267, USA; 2Cancer Research Scholars Program, College of Allied Health Sciences, University of Cincinnati, Cincinnati, OH 45267, USA; 3Simmons Comprehensive Cancer Center, University of Texas Southwestern Medical Center, 5323 Harry Hines Boulevard, Dallas, TX 75390, USA

**Keywords:** HER3, alteration, bladder cancer, breast cancer, colorectal cancer, lung cancer

## Abstract

**Simple Summary:**

HER3 is a member of the HER family. In the past decade, HER3 has been recognized as an important protein in allowing tumors to grow unchecked. This review focuses on HER3′s involvement in bladder, breast, colorectal, and lung cancers. In addition, this review covers what is known about HER3 alterations or mutations in bladder, breast, colon, and lung cancers. Finally, therapies that target HER3 are discussed.

**Abstract:**

In recent years, the third member of the HER family, kinase impaired HER3, has become a target of interest in cancer as there is accumulating evidence that HER3 plays a role in tumor growth and progression. This review focuses on HER3 activation in bladder, breast, colorectal, and lung cancer disease progression. HER3 mutations occur at a rate up to ~10% of tumors dependent on the tumor type. With patient tumors routinely sequenced for gene alterations in recent years, we have focused on HER3 mutations in bladder, breast, colon, and lung cancers particularly in response to targeted therapies and the potential to become a resistance mechanism. There are currently several HER3 targeting drugs in the pipeline, possibly improving outcomes for cancer patients with tumors containing HER3 activation and/or alterations.

## 1. Introduction

HER3 (ERBB3) is a member of the HER (ERBB) family, along with EGFR (HER1, ERBB1), HER2 (ERBB2) and HER4 (ERBB4). This group of receptor tyrosine kinases (RTKs) is involved in cell growth, signaling, and survival [1]. HER proteins are composed of an extracellular domain (ECD), a single transmembrane domain and an intracellular domain where the tyrosine kinase domain is found (Figure 1a). The ECD is further divided into subdomains I, II, III and IV. In the inactive state HER family members exist in a closed or tethered conformation where there are interactions between ECD subdomains II and IV, preventing homo- or hetero-dimerization with other HER family members [2]. Ligand binding occurs at ECD subdomains I and III, resulting in a conformational change to the open or extended form, revealing a dimerization arm [3]. HER proteins in the open state can dimerize with another HER protein resulting in further conformational changes in the intracellular domain. Subsequent phosphorylation events involving one monomer acting as the ‘donor’ in which the C-lobe (c-terminal portion of the kinase domain) interacts with the N-lobe (n-terminal portion of the kinase domain) of another monomer ‘acceptor’ [4]. The donor receptor can initiate phosphorylation of tyrosine residues in the C-terminal tail of the acceptor receptor. These phosphorylated tyrosine residues serving as docking sites for adaptor proteins and result in activation of downstream signaling (Figure 1a) [5].

EGFR has multiple ligands including high affinity ligands: epidermal growth factor (EGF), transforming growth factor-alpha (TGFα), heparin-binding EGF-like growth factor (HB-EGF), betacellulin (BTC) and lower affinity ligands: amphiregulin (AREG), epiregulin (EREG), and epigen (EPGN) [6]. HER3 and HER4 ligands are neuregulins (heregulins) NRG 1-4 [7,8]. HER2 has no known high affinity ligand and structural studies reveal that HER2 is always in the open conformation, poised to dimerize with other ERBB family members [9]. An assortment of ligands and dimerization partners allows for a range of downstream signaling pathway activation [10,11].

HER3 has long been considered kinase dead due to mutations in conserved Asp813 and Glu738 [2,3,4,5]. There is evidence that HER3 has some kinase activity (500-fold less kinase activity than EGFR [12], but it has not been demonstrated that this activity is required for HER3-dependent functions. To activate downstream pathways, HER3 must dimerize with another RTK to form a heterodimer. As HER2 has the strongest catalytic kinase activity of all HER family members [13], there is evidence that HER2-HER3 dimers form the strongest signaling dimer considering all potential combinations [14]. Recently, the published HER2-HER3-NRG1β structure determined by cryo-electron microscopy found a complex dynamic interface enabling allosteric sensing of ligand occupancy [15]. Another preferred dimerization partner of HER3 is EGFR, whereas HER3-HER4 dimers form at a lower affinity [11]. In addition, HER3 can dimerize with non-HER family members, including the FGFR family, MET, and Axl [16,17,18,19]. Strikingly, HER3 is the only ERBB family member that can directly activate PI3K/Akt, as there are 6 HER3 phosphorylation sites that the p85 regulatory subunit of PI3K can bind to [20,21]. EGFR and HER2 can activate the PI3K/Akt indirectly with adaptor molecules GAB1 and GRB2 [22] and all HER family member dimers can activate the MAPK pathway with GRB2 and SHC adaptors [23]. Taken together HER3 can activate the PI3K/Akt pathway, which is significant for cancer cell survival and metabolism [24] and the MAPK pathway, inciting cell proliferation [25].

## 2. Incidence of HER3 Mutations in Human Cancers

With the advent of high throughput massively parallel sequencing technology of next generation sequencing, the first major study to explore the role of HER3 mutations in cancer was Jaiswal et al. in 2013, in which the frequent occurrence in colon and gastric cancers was studied [26]. It was found that when mutant HER3 was expressed in conjunction with HER2, downstream levels of p-HER3, p-HER2, p-Akt, and p-ERK showed an increase and this resulted in anchorage dependent growth in both breast and colonic cells. This evidence and the frequency of HER3 mutations showed that mutant HER3 is potentially oncogenic for further exploration [27,28]. Later studies showed that certain HER3 mutations were found to be oncogenic, and in recent years, further papers have focused on the role mutant HER3 and other HER3 alterations play in different types of cancers [26,29,30]. This review will focus on the role of HER3 in bladder, breast, colorectal, and lung cancers including HER3 alterations, as HER3 alterations are frequently found in these subsets (Figure 2), and discuss the top HER3 mutations seen across each tumor type (Figure 1b) [27,28].

### 2.1. Bladder Cancer

Bladder cancer is a disease in which uncontrollable cell proliferation results in the growth of tumors that impair bladder function, particularly in the urothelial cells and includes tumors of the bladder, upper urinary tract, and proximal urethra [31]. There are two main subgroups of bladder cancer: non-muscle invasive bladder cancer (NMIBC), which does not typically threaten survival prognosis but nearly always recurs, and muscle invasive bladder cancer (MIBC), which are typically fatal due to the fast rate of metastatic progression [32]. The 5-year recurrence-free rate also depends on the patient’s risk for microscopic hematuria, with low and intermediate being 43% and 33%, respectively, and high being 21% [33].

The treatment regimen is determined by the site and stage of the disease in addition to patient risk factor (low, intermediate, high) for microscopic hematuria. Typically, the primary treatment regimen is surgery plus neoadjuvant chemotherapy, with the chemotherapeutics of choice being cisplatin, methotrexate, vinblastine, doxorubicin, and gemcitabine. Typically, the surgery is the transurethral resection of bladder tumor (TURBT), which is conducted for non-metastatic forms of bladder cancer. In low or intermediate risk patients, a single dose of intravesicle chemotherapy may be administered within 24 h of TURBT [33]. For those with muscle-invasive but localized urothelial carcinoma, patients are treated with neoadjuvant chemotherapy followed with radical cystectomy, pelvic lymph node dissection, and urinary diversion or another form of bladder-sparing treatment [34]. 

Depending on the receptors that are co-expressed, the presence of HER3 could be either a favorable or unfavorable prognosis and more studies are required to determine its role. Nielson et al. found that when co-expressed with MUC1, the presence of HER3 resulted in a favorable prognosis [35], whereas studies described below point to HER3 acting as an oncogene. It has been noted by Memon et al. that higher soluble HER3 expression is tied to a more favorable outcome and soluble HER3 inhibits bladder cancer activation of HER3, cell growth and migration [36]. 

Weickhardt et al. observed that the increased expression of phosphorylated HER3 is a key mechanism in the development of resistance to FGFR inhibitors in FGFR3 fused bladder cancers [37]. It is known that FGFR3 alterations lead to oncogenic signaling through the MAPK and PI3K pathways and that these alterations are particularly sensitive to FGFR inhibitors, leading to the approval of erdafitinib for patients with advanced bladder cancer [38]. However, patients developed resistance quickly and a study investigated epithelial to mesenchymal transition, EGFR, HER2, and HER3 activation as resistance mechanisms in the RT-112 cell line [37]. Weickhardt et al. treated two bladder cell lines harboring FGFR3 fusions (RT4 and SW780) with multiple FGFR inhibitors and saw an increase in phosphorylated HER family members, particularly HER3, after treatment with the FGFR inhibitor BGJ398 [37]. Given the data collected from this study, it’s possible that HER3 alterations could play a role in drug resistance. Based on data indicating HER2 and HER3 mutations are associated with efficacy of immune checkpoint inhibitors [39], a patient with metastatic urothelial carcinoma (mUC) was treated with anti-PD-1 antibodies and paclitaxel even though the tumor was PD-1 negative but contained an in-frame insertion of HER2 and HER3 amplification [40]. After two cycles of treatment, the lung metastases had significantly shrunk, and the patient continued to exhibit a partial response and progression-free survival for up to 19 months. This study points to the potential utility of HER3 alterations as a biomarker in predicting response to immune checkpoint inhibitors.

The most common HER3 alteration in bladder cancer is mutation, with 10% of all bladder cancers across 16 non-redundant studies in cBioPortal harboring a HER3 mutation (n = 2419) (Figure 2) [27,28]. The most common mutation is M91I, the biological significance of which is unknown; however, it has been denoted as a statistically significant hotspot. The second most common HER3 mutation seen in bladder cancer is V104M/L/Wfs*18, where the valine (V) at the 104th amino acid position is changed to either a methionine (M) or leucine (L) or involved in a frame shift deletion (Figure 3). Out of these three, the M/L mutation is known (M) or likely (L) to be oncogenic [26,29,41,42,43,44]. 

HER3 and co-occurring mutations have been the scrutiny of targeted therapies. Milewska et al. explored the effect that PI3K inhibitors have on HER family and PIK3CA mutated cancers [45]. Somatic HER3 mutations may convey sensitivity to certain treatments. Tamura et al. examined how HER2 and HER3 mutations conveyed sensitivity to the targeted therapies dacomitinib (PIK3CA) and afatinib, an irreversible HER family inhibitor, in bladder cancer via in vitro methods [46]. Choudhury et al. further explored the effect HER family mutations had on the effectiveness of afatinib in urethral carcinoma patients [47]. It was found that with afatinib treatment the median time to disease progression was 6.6 months in patients with HER2 or HER3 alterations in comparison to 1.4 months in patients without alterations. These data show that HER3 alterations may affect patient sensitivity to targeted therapy and that the presence of HER3 alterations could be a biomarker for afatinib effectiveness in bladder cancer.

### 2.2. Breast Cancer

From a histopathological perspective, breast cancer comprises three main subtypes: hormone receptor positive (estrogen receptor+ [ER+], progesterone receptor+ [PR+]), HER2-positive (HER2+), and triple negative breast cancer (TNBC) [48]. These subtypes determine the treatment regimen that is used for patients and are the basis for overall prognosis. Hormone receptor positive breast cancer is the most common form of the disease. Comprising roughly 16% of all breast cancers, triple negative breast cancer lacks the expression of ER, PR, or HER2 and has a high heterogenous nature between patients [49]. These patients have a poorer prognosis with fewer treatment options. Although there have been recent advances in triple negative breast cancer including approved treatments targeting human trophoblast cell-surface antigen 2 (Trop-2), systemic therapies, and immunotherapy [50,51,52].

HER3 is an important molecule in hormone positive or luminal breast cancers. HER3 mRNA is highest in ER+ or luminal tumors [53,54] consistent with the observation that HER3 is required for cell survival in the luminal but not the basal normal mammary epithelium [55]. Treatment with the ER down regulator fulvestrant induces protein expression and activity of HER3 in ER+ breast cancer cells in vitro. This upregulation of HER3 drives neuregulin-mediated resistance to fulvestrant [56]. Another study demonstrated that fulvestrant-resistant MCF-7 cells depend on increased HER3 and neuregulin-2 expression to maintain their growth and survival [57]. Moreover, treatment of primary breast tumors with fulvestrant [58] results in increased HER3 expression and PI3K/mTOR signaling [54]. The use of a HER3 neutralizing antibody in fulvestrant-treated tumor cells decreased tumor cell growth [54]. Recently activation of HER3 in luminal breast cancers has been shown to be mediated in a paracrine manner by cancer-associated fibroblasts (CAFs) expressing HER3′s ligand NRG1 [59]. The NRG1 ligand has also been implicated in endocrine resistance in ER+ breast cancers [60].

HER3 plays a critical role in HER2-mediated tumorigenesis. Tumors occurring in mice overexpressing the Neu transgene, the rat homolog of human HER2, exhibit increased expression and phosphorylation of HER3 [61]. Loss of HER3 causes a decrease in cell proliferation related to loss of signaling through PI3K, suggesting that HER2 in the absence of HER3 is insufficient to promote breast cancer cell proliferation [62]. siRNA-mediated knock-down of HER3 in HER2-overexpressing breast cancer cells reduces phosphorylation of HER2 and cell growth [63]. HER3 is as essential as HER2 for maintaining cell viability in HER2-overexpressing breast cancer cells [64]. Loss of HER3 prevents HER2-mediated transformation of mammary epithelium in transgenic mice [65]. HER2 TKIs impair HER3 phosphorylation and its subsequent interaction with PI3K in HER2-overexpressing cells [66,67].

Evidence has shown that HER3 may play a role in brain metastasis, as HER3 has been observed to be present in breast cancer brain metastases [68]. It was noted by Kodack et al. that there was an increase in HER3 expression in brain metastases and had higher phosphorylation levels in HER2 amplified and PIK3CA mutated cancers, even after treatment with drugs targeting those molecules [69]. HER2+ breast cancer metastasis and degree of treatment effectiveness. HER2 is the preferred dimerization partner of HER3, interacting in a powerful heterodimer with interactions at the kinase interface [14,70]. Inhibition of the HER2 signaling pathway results in upregulation of HER3 RNA and protein [71,72,73] A study by Al-Akhrass et al. indicated that a feed-forward loop between SorLA and HER3 determines heregulin response and neratinib resistance via the upregulation of HER2 and HER3 [74] Furthermore, it has been demonstrated that treating HER2+ breast cancer cells with the HER2 tyrosine kinase inhibitor lapatinib results in higher HER2/HER3 heterodimerization than prior to treatment [75]. Based on previous data showing HER2+ breast cancer patients having a higher chance of developing brain metastases, Lim et al. explored the landscape of HER family heterodimers in a cohort of breast cancer patients with brain metastasis [76]. It was found that brain metastases had significantly higher incidences of the HER2-HER3 heterodimer than the primary tumor. These data indicate that the HER2-HER3 heterodimer plays an important role in brain metastasis, potentially having dire consequences in terms of clinical outcomes and treatment regiments, as many existing targeted therapies for HER2+ cancers, such as large molecules like antibodies, have a limited ability to cross the blood–brain barrier (BBB). 

There is evidence that HER3 plays a role in TNBC [77]. Miano et al. found that the HER3/HER2 axis triggered the anchorage-independent growth of basal-like and triple-negative breast cancer cells when activated by neuregulin (NRG1) [78]. Ogden et al. found that the EGFR-HER3 heterodimer led to decreased survival in triple negative breast cancer in comparison to the overexpression of either receptor alone [79]. This dimer led to an increase in EGFR and PARP1 activation and could result in therapeutic resistance. Lyu et al. recapitulated these studies, finding that HER3 was upregulated after treating TNBC cells with the histone deacetylase inhibitor Panobinostat, suggesting that HER3 signaling acts as a resistance mechanism [80].

HER3 amplification in breast cancer has been explored more thoroughly in recent years and has been implicated as a possible prognostic marker. Previous studies indicated that there was some form of crosstalk between HER2 and HER3, particularly in breast cancer patients [81]. Li et al. found that the upregulation of HER3 via HER2 3′-UTR mediation may be involved in breast cell transformation, increased tumor growth, and potential resistance to HER2 targeted therapy [82]. Another study by Buccina et al. observed that HER3 signaling could be targeted using the HER2 inhibitor, lapatinib, in combination with the AMPK activator GSK-621 [83]. In this study, it was seen that when GSK-621 was used in combination with lapatinb, it suppressed NRG-regulated up-regulation of HER3 signaling in HER2+ breast cancer. Additionally, the GSK-621/lapatinb combination decreased p-HER2 levels; however, p-HER3 levels were not affected by the drug combination.

The most common type of HER3 alteration in breast cancer is mutation. Data reported across multiple cohorts (n = 8521 samples) in cBioPortal indicate that about 2% of all breast cancers harbor a HER3 mutation (Figure 4) [27,28]. The most common HER3 mutation found in breast cancer is E928G followed by V104L, G284R, and T355I. Our laboratory examined the role mutant HER3 plays in ER+ and HER2+ breast cancer [84]. 8 different patient derived mutants were examined, and out of those studied, the T355I variant was found to be the most oncogenic activating the MAPK pathway. Modeling studies found that mutating amino acid threonine 355 to the bulkier more hydrophobic isoleucine residue may disrupt a polar pocket present in the closed or tethered conformation. This could lead to a conformational shift to the extended, active form of HER3, even without ligand present. 

Clinical studies have shown that HER3 mutational status has an impact on patient response to therapy. Bidard et al. studied a case where a patient with ER+, PR+, HER2- invasive ductal carcinoma that had metastasized to the liver and conducted whole genome sequencing before treatment was administered [85]. The results showed that an activating, oncogenic G284R HER3 mutation was present, confirmed by Sanger sequencing. After surgery and administration of first- and second-line therapy, doxorubicin and docetaxel, respectively, disease progression was observed at the 11-month mark and the patient was then treated with a combination of two HER2 targeting therapies, trastuzumab and lapatinib. A partial response according to RECIST criteria was seen after 8 weeks of treatment and was sustained for the 40 weeks of follow up [85]. This data demonstrates that HER3 mutational status could play a role in how a patient responds to HER2 therapy.

Additionally, Hanker et al. showed that co-occurring HER2 and HER3 mutations cooperate to activate the PI3K pathway in different cancer models [86]. HER2-HER3 heterodimers have potent catalytic activity. This activity is compounded by the presence of both HER2 and HER3 mutations, leading to the higher activation of HER2 and increased catalytic activity. Additionally, it was explored whether HER3 mutational status contributed to drug sensitivity. Dimerization simulations were conducted using HER3 WT, HER3 E928G, HER2 WT, and several different HER2 mutants and compared with different HER2 targeted therapies, particularly neratinib. It was found that the HER3 E928G mutant modulated sensitivity to neratinib, poziotinib, afatinib, and tucatinib, which indicates that this mutation could act as a resistance mechanism for HER2 targeted therapies.

### 2.3. Colorectal Cancer

Colorectal cancer (CRC) is the third leading cause of cancer deaths in the United States, with 151,030 estimated new cases and 52,580 estimated deaths in 2022 [87]. Slightly more common in men and older populations, the overall 5-year survival rate for colon cancer is 63% [87]. There are several different subtypes of colorectal cancer, with the first subsets being classified between the colon and the rectum. These two subsets are further classified by the type of tissue in which the disease originated. In the scope of HER3 contribution to cancer progression, most studies have been focused on HER3 amplification as a therapeutic target; however, in recent years mutant HER3 has become an area of interest. 

There are three main HER3 alterations seen in colorectal cancer patients: amplification, deletion, and mutation (Figure 2). 2.86% of colon cancers have an ERBB3 amplification [27,28]. In these cases, there is an upregulation of HER3 and downstream pathways may have increased activation. Several papers have explored the effect higher HER3 levels have on the prognosis and progression of disease in colorectal cancer, particularly in the context of metastasis. Lédel et al. examined 236 patients with stage II and III colorectal cancer. There was a worse prognosis for patients with higher HER3 levels as assessed by immunohistochemistry [88]. Styczen et al. observed that 75% of patients with colorectal liver metastasis (n = 208 total patients examined) had high (IHC score 2+ or 3+) HER3 levels [89]. These findings were recapitulated by Rathore et al. when they found that the liver endothelium promotes HER3/Akt signaling by paracrine ligand activation [90]. 

Although less common than ERBB3 amplification, ERBB3 deletion (0.04% frequency) could play a role in how cancer progresses (Figure 2) [27,28]. It was noted by Lee et al. that ERBB3 deletion in a mouse intestinal model resulted in tumor specific apoptosis [91]. This finding also posed the possibility that HER3 plays a role in tumorigenesis and disease progression. Additionally, Rojas et al.’s study involving ERBB3 deletions revealed that cancers had increased EGFR expression when HER3 was not present and that tumors with ERBB3 deleted were smaller than those with ERBB3 present [92].

Although endogenous ERBB3 deletions have not been widely explored, it has been observed in knockout mouse models that the deletion of HER3 led to the reduction in proliferation of colon polyps [91,92]. Additionally, it was shown by Lee et al. that by knocking down HER3 in HCT116 colon cancer cells, cell cycle arrest was induced in addition to the activation of Bax-Bak dependent apoptosis [93]. Further exploration in naturally occurring ERBB3 deletion is needed to explore potential effects on oncogenic signaling and growth.

Across all colorectal cancer data sets in cBioPortal (n = 4453), an average of 5% of all colorectal cancers harbor an ERBB3 mutation, with several of these being considered hotspot mutations [27,28]. The Firehose Legacy data set, containing 640 patient samples, reports a 6% HER3 mutation rate and the location of mutation within the receptor could potentially influence the level of activation in downstream molecules, as certain HER3 mutations have been defined as oncogenic hotspots. Unlike with other oncogenes, such as BRAF and KRAS, there is no overall predominant mutation of HER3 and the most common variant differs between tumor types [30]. The most common mutation in colorectal cancer is the V104M mutation, which occurs in ECD I (Figure 5). In this mutation, the hydrophobic valine (V) is mutated to hydrophobic, bulkier methionine (M). Additionally, this mutation has been defined as a hotspot mutation and is the most common seen in the cBioPortal metastatic cohort, hinting that this could be involved in metastatic potential. The A232V/T is the second most common, occurring in extracellular domain II and the alanine (A) is changed to a valine (V) or threonine (T), and is known to be oncogenic, potentially making it a future biomarker for targeted therapies [26,29,30]. 

Although drugs targeting HER3 have not been approved for use in colorectal cancer, EGFR targeted therapy is approved such as cetuximab. Additionally, as HER3 is kinase impaired, having 500-fold less kinase activity than other members of the HER family [12] it is possible to take advantage of these already approved targeted therapies to prevent downstream signaling. However, HER3 could possibly act as a bypass and activate other pathways to encourage cell growth, survival, and motility. It was observed by Bosch-Vilaró et al. that HER3 dimerization with HER2 was induced in response to in vitro treatment with cetuximab, providing an escape mechanism to this FDA-approved targeted therapy [94]. 

HER2, a binding partner of HER3 that forms a potent heterodimer, has become a target of interest in colorectal cancer clinical trials, as HER2 amplification is found in 5% of all colorectal cancers. Already a target in other cancers, such as breast and gastric, HER2 alterations have been begun to be explored in colorectal cancer patients [95]. However, HER3 could potentially act as a bypass mechanism for HER2 targeted therapies. Ross et al. explored the genomic landscape of colorectal cancer, particularly in the context of HER2 and HER3 mutations [95]. It was later established by Hanker et al. that the HER3 E928G mutation resulted in residual phosphorylation of the Akt pathway after administration of HER2 targeted therapies (i.e., neratinib, tucatinib, lapatinib) [86]. Although this was in the context of co-occurring HER2 and HER3 mutations, the data suggests that mutant HER3 could provide a resistance mechanism to HER2 targeting therapies.

Currently, the standard of care for colorectal cancer patients is a combination of surgery and adjuvant chemotherapy [96]. One of the main focuses of targeted treatments in colorectal cancer is EGFR, which upon homo- or hetero-dimerization with another RTK to activate downstream signaling pathways. Although HER2 is HER3′s preferred dimerization partner, the commonality of HER3 amplifications and mutations may leave it with a higher possibility of dimerizing with EGFR [97]. Given this information, this raises the possibility that the HER3 pathway may act as an escape mechanism. However, in some studies, HER3 amplification has been implicated as a favorable prognostic marker for patients. In Seligmann et al.’s study of patients in a phase III trial of panitumumab, a monoclonal antibody targeting EGFR, it was found that high HER3 mRNA expression correlated to a better OS but shorter progression free survival (PFS) [98]. 

HER3 treatments can be used in combination with methods of standard treatments, such as surgery and systemic chemotherapy. Rathore et al. noted that a combination of 5-FU, one of the standards of care in metastatic CRC treatment, and seribantumab, a HER3 targeting monoclonal antibody, had a greater effect in inhibiting tumor growth in vivo than either treatment alone [90]. Irinotecan, another widely used chemotherapeutic used in CRC treatment, was tested as a potential combination with seribantumab; however, it was determined that the toxicity resulting from this combination was too great for patients to tolerate. It has also been noted in Hanker et al. that a combination therapy can influence cell proliferation and survival [86]. In this study, colon cancer cells were treated with a combination of the pan-HER inhibitor neratinib and PI3K inhibitor alpelisib, resulting in the elimination of pAkt and reduction in cell proliferation and survival.

### 2.4. Lung Cancer

Lung cancer is the leading cause of cancer deaths worldwide and non-small cell lung cancer (NSCLC) is the deadliest subtype among this group [87]. Currently, there are several targeted therapies approved to treat lung cancer, with the targets of interest being ALK, MET, HER2, and EGFR [99]. As all these receptors could potentially be involved in crosstalk with HER3. Alterations in HER3 may influence targeted treatments aimed towards these receptors.

HER3 has been implicated as a possible therapeutic target in lung adenocarcinoma. Although mutant EGFR is more common and has approved therapeutic treatments, HER3 is able to dimerize with it and propagate downstream signaling as a type of resistance mechanism [100]. It was shown by Manickavasagar et al. that patients with EGFR mutation tended to co-express HER3 and although there was not a significant difference between the two groups, this could be due to small sample size [101]. Additionally, it was observed that MEK activity was increased when HER3 expression was high, indicating that HER3 may provide an escape mechanism to EGFR therapies. It was first noted by Yi et al. in 1997 that high cytoplasmic HER3 expression correlated with shorter survival in advanced non-small cell lung carcinomas [102]. Later, Muller-Tidow et al. found that HER3 overexpression was associated with decreased survival time in lung cancer patients, and a further study by Ma et al. recapitulated these previous findings by noting that the overexpression of HER3 was observed in lung cancer cells but not normal lung cells [103,104]. Furthermore, the gene that codes for HER3, ERBB3, has been included as both part of a two gene signature with BRCA1 in which both genes are regarded as independent risk factors in the IASLC/ATS/ERS classification of early-stage lung adenocarcinoma and a five gene signature that is predictor of OS and relapse-free survival (RFS) in NSCLC patients [105,106].

In more recent years, HER3 expression has also been explored in relation to brain metastases [107,108]. Scharpenseel et al. examined EGFR and HER3 expression in the primary tumors and their brain metastases in lung cancer patients to see if overexpression played a role in disease progression and metastasis [109]. Additionally, they looked at circulating tumor cells (CTCs) and found that there was a high co-expression of EGFR and HER3. Based on these data, they were able to detect CTCs based on HER3 and EGFR expression via magnetic cell separation. This study could provide the basis for the use of HER3 overexpression in combination with other biomarkers to tailor a patient’s treatment regimen to their tumor’s genetic makeup and track disease progression.

To enhance the use of biomarkers in the diagnostic process, Masroor et al. explored the significance of ERBB3 and ERBB4 mRNA in the prognosis of lung adenocarcinoma [110]. It was found that patients both with and without distant metastases had a significant difference in ERBB3 mRNA expression with metastatic patients having higher HER3 expression. Additionally, data showed that patients with a ERBB3 mRNA expression >8 fold had an average OS of 5.45 months shorter than that of those with lower mRNA expression (OS = 10 months). This decrease in the OS and increase of metastases in lung adenocarcinoma patients with higher HER3 mRNA expression suggests that HER3 plays a role in disease progression and metastasis development.

Kawano et al. correlated ERBB3 mRNA expression to pathological features of lung cancer [111]. A cohort of 192 NSCLC patients that underwent surgery as primary treatment were sorted into different categories based on their clinical and pathological characteristics and their tumors examined for ERBB3 expression using RT-PCR. It was found that an increased expression of ERBB mRNA may play a role in NSCLC, particularly in females, non-smokers, adenocarcinoma patients, and those with an EGFR mutation. To corroborate these findings, immunohistochemistry was conducted on tumors samples to determine HER3 protein expression. However, it was found that HER3 expression was higher in smokers, although there was not a difference in prognosis between HER3+ and HER3- groups. Potentially, these findings could point to if HER3 status modulates patient response to EGFR targeted therapy.

Drug tolerance and resistance to targeted therapy has been an increasing problem for lung cancer patients. Tanimura et al. explored the contribution of HER3 to the rise of ALK-inhibitor tolerant cells, particularly those with mesenchymal features [112]. Taniguchi et al. explored this phenomenon further by generating two cell lines resistant to lorlatinib, a 3rd generation ALK inhibitor, and found that they overexpressed phosphorylated HER3 [113]. Additionally, no ALK mutations were found and cross-resistance to other ALK inhibitors (i.e., crizotinib and alectinib) was observed. These data show that HER3 potentially plays a role in drug resistance to targeted therapies in lung cancers.

HER3 has been shown to be a possible therapeutic target when used in combination with other forms of treatment. Toulany et al. explored the effect of reducing nuclear Akt activation by targeting HER3 and combining it with radiotherapy [114]. It was observed that after treating NSCLC cell lines with ionizing radiation, the phosphorylation of Akt was immediately slightly increased at phosphorylation sites S473 and T308. The same cell lines were then transfected with small interfering RNA (siRNA) targeting EGFR, HER2, and HER3 followed by ionizing radiation. It was found that out of these three HER family members, HER2 and HER3 knockdown resulted in diminished post-irradiation induced activation of Akt [114]. This demonstrates that targeting of HER3 may have potential positive clinical implications when used in conjugation with other forms of therapy. Additionally, in Romaniello et al.’s study, it was found that after treatment with osimertinib, a 3rd generation EGFR TKI, there was an upregulation of HER3 that contributed to drug resistance [115]. Treatment with an anti-HER3 antibody (mAb33) and EGFR targeted therapy (osimertinib and cetuximab) showed that HER3 targeted treatment in combination with other regimens may be an effective second line therapy for those that develop resistance to EGFR targeted therapy. Additionally, a HER2 targeting monoclonal antibody (trastuzumab) was also tested in combination with cetuximab and the same upregulation of HER3 was observed. These data support the targeting of HER3 in lung cancer to prevent the development of a resistance mechanism to therapies targeting other members of the HER family in lung cancer [115].

Although there are naturally occurring HER3 amplifications and deletions, the most common form of HER3 alteration in lung cancer is mutation, with the highest rate of frequency mutation in lung cancer being E928G (Figure 6) [27,28]. This mutation is in the kinase domain and known to be oncogenic. It was shown by Littlefield et al. that this mutation increases the activator function of EGFR and given the role EGFR plays in the progression of lung cancer, it is possible that mutant HER3 could be a resistance mechanism to EGFR targeted therapies [116,117]. 

A case study found that a novel HER3 mutation contributed to chemotherapy resistance in an adolescent patient with advanced non-small cell lung cancer [118]. The data showed that the HER3 V855A in the kinase domain was at a position homologous to the known EGFR L858R driver mutation. As HER3 contains six binding sites for the PI3K p85 subunit that are absent in EGFR or HER2, the conclusion was drawn that HER3 is a strong point for PI3K/Akt signaling [119,120]. This demonstrates that certain HER3 mutations could have activity homologous to other members of the HER family when active.

It was previously established that HER3 germline mutations could potentially lead to familial cancer predisposition, particularly in erythroid MDS/erythroleukemia, but this also translates to other forms of cancer [121]. McInerney-Leo et al. published a case study investigating the role of a germline HER3 mutation in hereditary lung adenocarcinoma [122]. Tumors from different members of a family were sequenced and found to be heterozygous for the same HER3 I649R mutation. It was found that a shorter form of HER3 (80 kDa) was produced at a higher ratio than that of the full-length form (185 kDa) when the novel mutant was present in addition to altered signaling pathways. 

One of the standards of treatment in lung cancer is targeting MET alterations. It has been shown that MET inhibition results in the mutation or amplification of other receptors, including HER3 [123]. This could potentially provide an escape mechanism to MET targeted therapy. Stern et al. explored the effect of MET amplification on the phosphorylation of HER3 and found that crosstalk between receptors encouraged cell proliferation and was mediated by MPZL3 [124]. This supports the hypothesis that HER3 could play a role in resistance to targeted therapies.

Other standards of care for lung cancer include inhibition of EGFR; however, resistance to treatment via mutation or the upregulation of other receptors is an ongoing problem [125,126]. In addition to EGFR mutations, an increase in HER3 expression has been implicated as one of these mechanisms, posing a problem as there are currently no approved HER3 inhibitors. In response to this need, Yonesaka et al. explored the combination of an EGFR small molecule inhibitor (osimertinib) and the HER3 antibody drug conjugate Patritumab-Deruxtecan (HER3-DXd) in patients with EGFR mutated cancer [127]. After examining 48 patient samples, it was found that after inhibiting the EGFR/Akt pathway, there was a noticeable increase in HER3 expression. To combat this phenomenon, the patient samples were treated with the HER3 antibody-drug conjugate HER3-DXd and Akt activation levels were evaluated. It was found that acquired EGFR-TKI resistance allowed for the enhanced efficacy of the antibody drug conjugate in vitro. This demonstrates that the combination therapy may be an effective second line treatment in the clinical setting.

## 3. Targeted HER3 Therapies

Currently, there are several HER3 targeted treatments that are in preclinical studies and clinical trials, which include monoclonal antibodies, multi-targeted antibodies, antibody-drug conjugates, small molecule inhibitors, and other HER3-targeted agents (Figure 7 and Table 1). 

HER3 targeting monoclonal antibodies (mAb) include Patritumab (U3-1287), Lumretuzumab (RG7116), GSK2849330, Seribantumab (MM121), LJM 716, and CDX-3379. Patritumab (U3-1287) is a fully human monoclonal immunoglobulin G1 (IgG1) antibody that binds to the ECD of HER3 to inhibit ligand interaction, resulting in the inhibition of downstream signaling along with a reduced HER3 expression [128]. Patritumab in combination with erlotinib had no dose-limiting toxicity in clinical trials [128,139,140]. Lumretuzumab (RG7116) is a humanized, glycoengineered immunoglobulin G1 (IgG1) antibody that binds to and inhibits HER3 by replacing the ligand neuregulin and locking the HER3 receptor in a closed conformation, therefore blocking downstream signaling. Lumretuzumab has not shown dose-limiting toxicity in clinical trials and can be used in combination with other inhibitors such as cetuximab or erlotinib [129,141]. GSK2849330 is a humanized IgG1/IgG3 monoclonal antibody (mAb) that shows similar specificity for the ECD of HER3 and blocks NRG binding [130]. It was used for patients with HER3-expressing cancer and no dose-limiting toxicity was observed; however, it limited the ability to block HER3-expressing tumors [130,142]. Seribantumab (MM121) is a fully human antibody that directly binds to the HER3 ECD and inhibits downstream HER3-dependent pathways via induction of HER3 receptor downregulation [131] and inhibits tumor development by blocking the PI3K-ATK pathway’s activation [131,142]. A phase II trial examined the combination of Seribantumab with Erlotinib or Erlotinib as a monotherapy in advanced, EGFR wild-type non-small-cell lung cancer who had not received any TKIs targeting EGFR. Although the primary endpoint of progression-free survival (PFS) in the intent-to-treat population was not met, seribantumab + erlotinib was associated with a significantly longer median PFS vs. erlotinib alone in the HRG positive group [143]. 

LJM716 is a fully human anti-HER3 immunoglobulin G1 (IgG1) monoclonal antibody. LJM716 is distinct from other HER2:HER3-targeted therapies in that it binds a conformational epitope that traps HER3 in the inactive conformation preventing its receptor activation and possesses the unique ability to inhibit both ligand-induced and ligand-independent activation of HER3. A phase 1 study evaluated the safety and tolerability of LJM716 in patients with HER2-positive breast cancer or gastric cancer, or with esophageal squamous cell carcinoma (ESCC) or squamous cell carcinoma of the head and neck (SCCHN) regardless of HER2 status [132]. This study reported that intravenously administered LJM716 was well tolerated, with an acceptable and manageable safety profile. A decrease in p-HER3/t-HER3 was observed in three out of five paired tumor biopsy samples after LJM716 treatment [132]. CDX-3379, an anti-ErbB3 monoclonal antibody, showed promising molecular and antitumor activity in head and neck squamous cell carcinoma (HNSCC) as a monotherapy or in combination with cetuximab. However, a phase II trial of CDX-3379 and cetuximab in patients with recurrent/metastatic cetuximab-resistant, HPV negative head and neck cancer reported a modest overall response rate with severe dose-limiting toxicities. This result impeded further clinical development of CDX-3379 and cetuximab as a combination [144].

Multi-targeted antibodies include Istiratumab (MM-141), Sym013, duligotuzumab (MEHD7954A), and MCLA-128. Istiratumab is a bispecific antibody that binds IGF-IR and HER3, prevents their ligands from binding and degrades the RTK complexes [133]. Sym013 is a recombinant pan-HER antibody mixture that targets EGFR, HER2, and HER3, effectively suppresses tumor growth, and inhibits the proliferation of cell lines in vitro [134]. However, Sym013 showed limited efficacy in patient groups, which included colorectal cancer [134]. Duligotuzumab (MEHD7945A) binds to both HER3 and EGFR, inhibiting signal transduction, leading to the inhibition of tumor proliferation. Duligotuzumab has a well-tolerated dose range of $30\frac{\mathrm{mg}}{\mathrm{kg}}$ and showed efficacious exposures to the patient group when given higher doses compared to lower doses [135]. MCLA128 targets both HER2 and HER3 receptors. Additionally, an increase in dose of MCLA128 was found to decrease the volume of the tumor in preclinical studies, demonstrating its efficacy as a potential therapeutic [136,145].

Antibody-drug conjugates (ADCs) are an up-and-coming class of drug in which a monoclonal antibody is covalently linked to a cytotoxic chemical [146]. Patritumab Deruxtecan (HER3-DXd) is an ADC comprised of a HER3 antibody linked to a topoisomerase I inhibitor. In preclinical studies, it showed efficacy in suppressing HER3 over-expression in a mouse model. A recent phase I clinical trial found that Patritumab Deruxtecan has clinical activity in EGFR TKI resistant NSCLC tumors. Strikingly 39% of the 57 patients receiving Patritumab Deruxtecan had an objective response rate. A wide range of HER3 membrane expressions was observed in the clinical activity and Patritumab Deruxtecan could be an approach to treat a broad range of drug-resistant tumors due to clinical activity in EGFR-TKI resistant cancers [137].

Efforts to develop small molecule tyrosine kinase inhibitors targeting HER3 have been few and far between as HER3 is kinase impaired, although several are in existence that target the pseudo kinase activity of HER3. TX1-85-1 is a small molecule that forms a covalent bond with Cys-721 located in the ATP-binding site of HER3. A derivative of TX1-85-1, TX-121-1, partially degrades HER3 to perturb HER3-mediated signaling [147]. A more recent approach in the use of a small molecule that targets HER3 has been to screen over 100,000 compounds and identify selective small molecule inhibitors that would trap HER3 in a conformation which is unfavorable for the formation of an active HER2–HER3 heterodimer. AC3573 was identified from this screen, as it bound to HER3 with some specificity and abrogated HER2–HER3 complex formation in cells along with downstream signaling [148].

Other agents targeting HER3 include designed toxins and antisense oligonucleotides. One such agent, Affitoxin-1, is a designed toxin that has a shortened version of diptheria toxin and HER3-binding affibody domains to form an affibody, a genetically modified small molecule that shows affinity for binding to target molecules. Afftoxin-1 induced apoptosis in HER3-expressing cells [149]. HER3 antisense oligonucleotide, EZN-3920 specifically downregulates the expression of HER3, corresponding to growth inhibition. There was improved efficacy when EZN-3920 was used with other anti-HER family drugs such as lapatinib, trastuzumab or PI3K inhibitors [150]. 

## 4. Conclusions

HER3 is a member of the HER family that has taken a backseat in research in comparison to EGFR and HER2. However, in more recent years, there is compelling evidence the role that HER3 plays in the disease progression of different cancer types, as HER3 has been implicated in disease progression and drug resistance in bladder, breast, colorectal, and lung cancer among others. In addition, studies regarding its altered forms have alerted researchers to the role HER3 mutations play in disease progression, prognosis, and treatment response/sensitivity. The receptor has become a promising target with tailored treatment options awaiting in preclinical and clinical trials, paving the way for potential new and exciting alternative treatments.

## Figures and Tables

**Figure 1 cancers-14-06174-f001:**
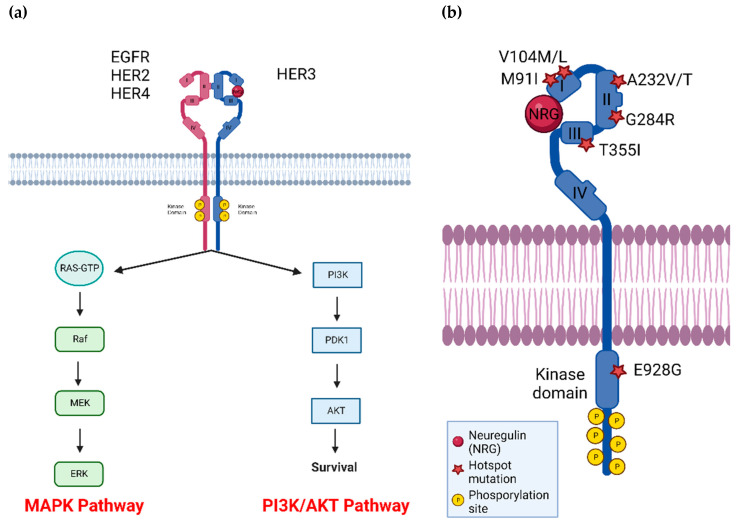
(**a**) HER3 dimerizes with other RTKs, most notably HER2, to activate downstream pathways involved in cell growth, survival, and signaling. (**b**) HER3 has several common mutations located across the four extracellular and kinase domains, as indicated by red stars. P: phosphorylation site, and NRG indicates the HER3 ligand neuregulin. Figure generated using BioRender.

**Figure 2 cancers-14-06174-f002:**
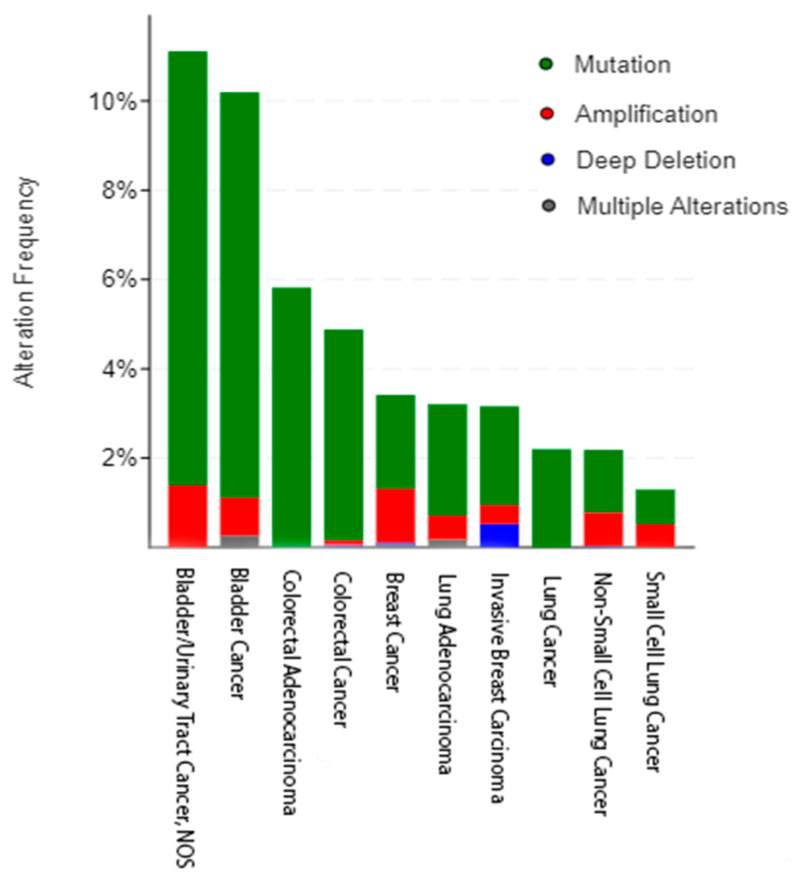
The frequency of HER3 alterations as seen in bladder (n = 2419), breast (n = 8615), colorectal (n = 4453), and lung (n = 7204) cancers across different subtypes and multiple nonredundant studies. Figure rendered using cBioPortal [27,28].

**Figure 3 cancers-14-06174-f003:**
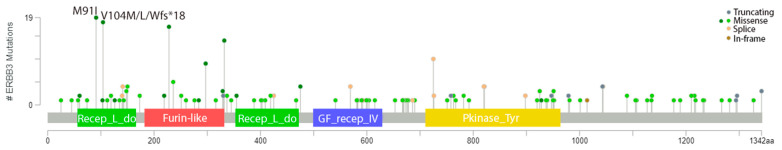
The frequency of HER3 mutations in bladder cancer and their locations within the receptor. Figure rendered in cBioPortal using multiple studies (n = 2419). Gray indicates truncating mutation, green indicates a missense mutation, yellow indicates splice mutation, and brown indicates an in-frame mutation [27,28].

**Figure 4 cancers-14-06174-f004:**
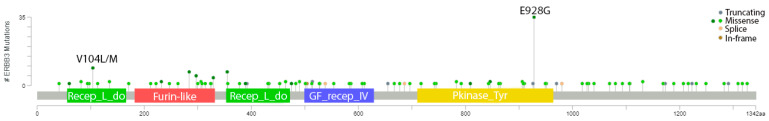
The frequency of HER3 mutations in breast cancer and their locations within the receptor. Figure rendered in cBioPortal using multiple studies (n = 8615). Gray indicates truncating mutation, green indicates a missense mutation, yellow indicates splice mutation, and brown indicates an in-frame mutation [27,28].

**Figure 5 cancers-14-06174-f005:**
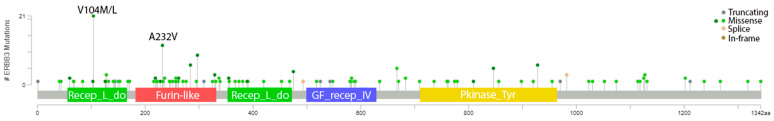
The frequency of HER3 mutations in colorectal cancer and their locations within the receptor. Figure rendered in cBioPortal using multiple studies (n = 4453). Gray indicates truncating mutation, green indicates a missense mutation, yellow indicates splice mutation, and brown indicates an in-frame mutation [27,28].

**Figure 6 cancers-14-06174-f006:**
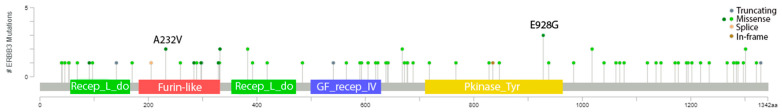
The frequency of HER3 mutations in lung cancer and their locations within the HER3 receptor. Figure rendered in cBioPortal using multiple studies (n = 7204). Gray indicates truncating mutation, green indicates a missense mutation, yellow indicates splice mutation, and brown indicates an in-frame mutation [27,28].

**Figure 7 cancers-14-06174-f007:**
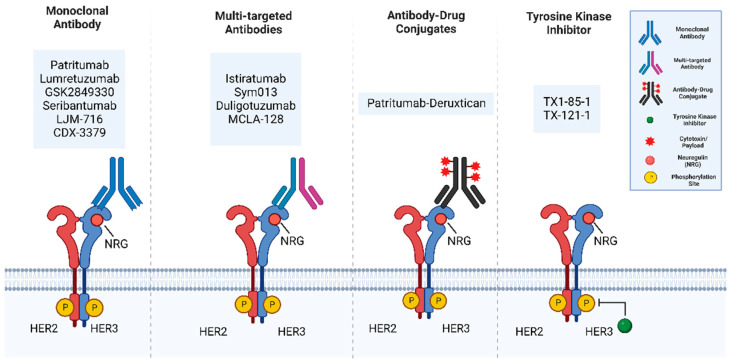
The main categories of HER3 targeting therapies include monoclonal antibodies, multitargeted antibodies, antibody-drug conjugates, and tyrosine kinase inhibitors (TKIs). Monoclonal antibodies bind to the extracellular domain of HER3. Multi-targeted antibodies include bispecific antibodies which combine two different antigen-binding sites in a single antibody. Antibody-Drug Conjugates are monoclonal antibodies chemically linked to a chemotherapeutic drug. Tyrosine kinase inhibitors are small molecules inhibitors that block tyrosine kinase activity. Figure generated using BioRender.

**Table 1 cancers-14-06174-t001:** Categories of HER3 targeting drugs and their mechanisms.

Category	Drug/Agent	Targeting Mechanism	Reference
Monoclonal Antibodies	Patritumab	IgG1 antibody that binds to ECD of HER3 to inhibit ligand interaction	[128]
	Lumretuzumab	Binds with high affinity to the ECD of HER3 to inhibits receptor activation	[129]
	GSK2849330	High specificity and affinity towards HER3 receptor. Blocks receptor dimerization and downstream signaling	[130]
	Seribantumab (MM-121)	Induces receptor downregulation and inhibition of downstream HER3-dependent signaling pathways.	[131]
	LJM716	Binds a conformational epitope that traps HER3 in the inactive conformation to prevent its receptor activation	[132]
Multi-targeting Antibodies	Istiratumab (MM-141)	Human tetravalent bispecific antibody that binds to/co-inhibits IGF-1R and HER3	[133]
	Sym013	Pan-HER antibody mixture. Binds specifically to non-overlapping epitopes on domain III of EGFR (Hu1277 and Hu1565), domain III and IV of HER2 (Hu4384 and Hu4517), and domain I of HER3 (Hu5038 and Hu5082)	[134]
	Duligotuzumab (MEHD7945A)	Humanized IgG1 antibody that targets both HER3 and EGFR	[135]
	MCLA-128	Humanized IgG1 bispecific targeting HER2 and HER3	[136]
Antibody-drug Conjugate	Patritumab-Deruxtecan	ADC comprised of an HER3 antibody linked with a topoisomerase I inhibitor	[137]
	EV20/MMAF	ADC composed of an HER3 antibody linked with a blocker tubulin polymerization.	[138]

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
