# Peer review of "HER3 Alterations in Cancer and Potential Clinical Implications"

_cancers, 2022, doi:10.3390/cancers14246174_

Round 1

Reviewer 1 Report

This is well constructed and interesting review. My main comment is that the figure legends could be improved:

1) Figure 1 - Label the domains. Include a key in the legend to define symbols including the star, P, NRG, colour code. 

2) Figure 3, 4, 5 and 6 - Increase the size so that the domain labels are readable. Include a key to the coloured dots in the figure legend. 

3) Figure 7 - Expand the legend. Include key to colour and symbol code.

Reviewer 2 Report

This article provides a comprehensive description of HER3, the molecule, its role in various cancers, and the utility of targeted therapy in these cancers. it may be helpful to understand that HER3 is a therapeutic target in cancer.
